# Multidrug-Resistant Avian Pathogenic *Escherichia coli* Strains and Association of Their Virulence Genes in Bangladesh

**DOI:** 10.3390/microorganisms8081135

**Published:** 2020-07-27

**Authors:** Otun Saha, M. Nazmul Hoque, Ovinu Kibria Islam, Md. Mizanur Rahaman, Munawar Sultana, M. Anwar Hossain

**Affiliations:** 1Department of Microbiology, University of Dhaka, Dhaka-1000, Bangladesh; otun.saha@gmail.com (O.S.); nazmul90@bsmrau.edu.bd (M.N.H.); okicpb@gmail.com (O.K.I.); razu002@du.ac.bd (M.M.R.); 2Department of Gynecology, Obstetrics and Reproductive Health, Bangabandhu Sheikh Mujibur Rahman Agricultural University, Gazipur-1706, Bangladesh; 3Department of Microbiology, Jashore University of Science and Technology, Jashore-7408, Bangladesh

**Keywords:** poultry, *Escherichia coli*, phylotypes, pathogenic, antibiotic resistance

## Abstract

The avian pathogenic *Escherichia coli* (APEC) strains are the chief etiology of colibacillosis worldwide. The present study investigated the circulating phylotypes, existence of virulence genes (VGs), and antimicrobial resistance (AMR) in 392 APEC isolates, obtained from 130 samples belonged to six farms using both phenotypic and PCR-based molecular approaches. Congo red binding (CRB) assay confirmed 174 APEC isolates which were segregated into ten, nine, and eight distinct genotypes by RAPD assay (discriminatory index, DI = 0.8707), BOX-PCR (DI = 0.8591) and ERIC-PCR (DI = 0.8371), respectively. The combination of three phylogenetic markers (*chu*A, *yja*A and DNA fragment TspE4.C2) classified APEC isolates into B2_3_ (37.36%), A1 (33.91%), D2 (11.49%), B2_2_ (9.20%), and B1 (8.05%) phylotypes. Majority of the APEC isolates (75–100%) harbored VGs (*ial*, *fim*H, *crl*, *pap*C, and *cjr*C). These VGs (*pap*C and *cjr*C) and phylotypes (D2 and B2) of APEC had significant (*p* = 0.004) association with colibacillosis. Phylogenetic analysis showed two distinct clades (clade A and clade B) of APEC, where clade A had 98–100% similarity with *E. coli* APEC O78 and *E. coli* EHEC strains, and clade B had closest relationship with *E. coli* O169:H41 strain. Interestingly, phylogroups B2 and D2 were found in the APEC strains of both clades, while the strains from phylogroups A1 and B1 were found in clade A only. In this study, 81.71% of the isolates were biofilm formers, and possessed plasmids of varying ranges (1.0 to 54 kb). *In vitro* antibiogram profiling revealed that 100% isolates were resistant to ≥3 antibiotics, of which 61.96%, 55.24%, 53.85%, 51.16% and 45.58% isolates in phylotypes B1, D2, B2_2_, B2_3_, and A1, respectively, were resistant to these antimicrobials. The resistance patterns varied among different phylotypes, notably in phylotype B2_2_, showing the highest resistance to ampicillin (90.91%), nalidixic acid (90.11%), tetracycline (83.72%), and nitrofurantoin (65.12%). Correspondence analysis also showed significant correlation among phylotypes with CRB (*p* = 0.008), biofilm formation (*p* = 0.02), drug resistance (*p* = 0.03), and VGs (*p* = 0.06). This report demonstrated that B2 and A1 phylotypes are dominantly circulating APEC phylotypes in Bangladesh; however, B2 and D2 are strongly associated with the pathogenicity. A high prevalence of antibiotic-resistant APEC strains from different phylotypes suggest the use of organic antimicrobial compounds, and/or metals, and the rotational use of antibiotics in poultry farms in Bangladesh.

## 1. Introduction

*Escherichia coli* is a ubiquitous organism having the fabulous adaptive ability in diverse ecological niches including the intestine of animals and humans [1]. This pathogen can also induce enteric and extraintestinal infections [1,2]. In particular, avian pathogenic *E. coli* (APEC) is the main causal agent colibacillosis in poultry farms; a syndrome associated with diarrhea and/or enteritis, mild to severe septicemia, airsacculitis, perihepatitis, and pericarditis [2]. However, in most of cases, the fundamental cause of the disease remains unclear, since the infection with *E. coli* is associated to the presence of *Mycoplasma gallisepticum* or respiratory viruses, such as Newcastle virus or Infectious Bronchitis virus [2]. Moreover, strains of *E. coli* also have zoonotic significance since they are known to cause infections both in humans and animals, including birds [3]. Currently, one of the greatest challenges for APEC is the difficulty of understanding their pathogenicity [4]. There are many approaches for molecular detection of various genomic fragments of the APEC, of them, polymerase chain reaction (PCR) is one of the best approaches, including studying their pathogenicity and/or virulence. The AFLP (amplified fragment length polymorphism) [5], RAPD (random amplified polymorphic DNA) [6], repetitive sequence-based PCR genomic fingerprinting [6], multilocus sequence typing (MLST) [7], BOX-PCR [8], Clermont phylotyping [9], and ERIC-PCR [10] are the most widely used PCR-based techniques for identifying APEC strains. Among the mentioned techniques, MLST has discriminatory power and reproducibility for typing *E. coli* [7], though it has some inherent limitations like relatively higher costs, unavailability and labor-intensiveness [7,11]. On the contrary, there are high correlations between the Clermont phylogenetic grouping and MLST analysis [9,11]. The *E. coli* strains can be grouped into different distinct phylogroups having originated from different ecological niches, and tendency to cause diseases in diverse host ranges including avian species [12,13,14,15,16]. Therefore, identifying and classifying the phylotype of an unknown strain can further expedite proper prevention and control programs, and also aid in designing rational treatment of infections caused by such strain [17], because of its simplicity, quickness and reproducibility [3,18]. The potentially pathogenic *E. coli* strains can be screened by different tests, like phenotypic assays of Congo red binding (CRB) [19]. Many researchers considered the CRB assay as an epidemiological marker to identify the APEC strains [20,21]. The binding of Congo red (CR) is associated with presence of virulence genes such as *omp*A, *iss*, *crl* and *fim*H, and genes for multiple resistance to antibiotics [20,21]. The functional amyloid fibers assembled by *E. coli* are called curli, and APEC are associated with curli production. As amyloid, curli fibers are protease resistant and bind to CR and other amyloid dyes [22,23]. In several previous studies, a strong correlation between CRB and pathogenic properties of APEC isolates was also reported [20,21,24,25]. The characteristic of CR binding constitutes a moderately stable, reproducible and easily distinguishable phenotypic marker, thus many researchers advocated the use of CR dye to distinguishing between pathogenic and non-pathogenic microorganisms in APEC study [20,21,24,25]. Several studies revealed the incidence of various phylotypes of APEC strains in combinations of virulence-associated genes (VGs) [7,26,27]. These virulence factors are associated with various virulence genetic markers, such as P fimbriae structural subunit (*pap*A) and P fimbriae assembly (*pap*C) [28], *fim*A (encoding type 1 fimbriae), bundle-forming pilus (*bfp*), aerobactin iron uptake system (*aer*) [29], *crl* (curli fimbriae), and many more which are linked to zoonotic concern [19]. Among all the existent adhesion factors, the P fimbriae is one of the essential factors in pathogenesis of the poultry epithelial cells [7]. The P fimbriae are important factors for the beginning and expansion of human urinary tract infections [30]; however, their role in the pathogenesis of avian colibacillosis is not yet clearly understood. The role of curli fimbriae which encodes for *crl* and *csg*A genes in the pathogenesis is poorly elucidated, though these genes facilitate the adherence of APEC strains to fibronectin and laminin [30,31,32].

Moreover, biofilm forming ability of the APEC strains is another vital virulence property [33] that justifies the reason for treatment failure using commercially available antimicrobials leading to persistence of the infections [34]. In addition, APEC isolates shared serotypes, VGs, and phylotypes with human diarrhoeagenic *E. coli* (DEC) isolates, which could subsequently be a potential public health concern [35,36]. Antibiotics resistance in bacteria is now a global public health concern, particularly the resistant property further aggravated due to indiscriminate use of antibiotics [37]. The relation of antibiotics resistance especially in APEC phylotypes has previously been discussed [31,38]. There are reports that resistant *E. coli* strains might enhance antimicrobial resistance in other organisms (pathogenic and nonpathogenic) within gastrointestinal tract of the chicken [3,39], and help in transmitting and disseminating drug-resistant strains, and genes between animal and human pathogens [3,39]. Many of the earlier studies have explored the association between pathogenic traits of APEC and their VGs repertoire [40,41,42,43]; however, potential pathogenic traits associated with phylotypes warrant attention into the potential role of VGs, MDR properties, and their cross-talk with specific phylotypes. This study; therefore, aims to investigate the phylotyping, potential VGs, and antimicrobial resistance (AMR) in APEC isolates. Furthermore, we study the association of APEC phylotypes in VGs carriage and AMR in the infected poultry samples of Bangladesh.

## 2. Materials and Methods 

### 2.1. Sample Collection and Processing

We collected 130 samples (healthy = 35; diseased = 95) including droppings (n = 30), cloacal swabs (*n* = 27), poultry feed (*n* = 11), handler’s swab (*n* = 9), egg surface swab (*n* = 10), feeding water (*n* = 10), and internal organ (liver, *n* = 33) from 6 commercial poultry farms belonging to Narsingdi (23.9193° N, 90.7176° E), Narayangonj (3.6238° N, 90.5000° E), and Manikgonj (23.8617° N, 90.0003° E) districts of Bangladesh. The study was conducted during April 2017 to March 2018 (Appendix A). Diseased chickens (those having colibacillosis) were confirmed by observing the clinical symptoms such as diarrhea and/or enteritis, mild to severe septicemia, airsacculitis, perihepatitis, and pericarditis [2]. Collected samples were kept into sterile plastic bags, carefully labeled, packed, cooled in icebox, transported subsequently to the laboratory, and stored at 4 °C. Further processing of all samples was done for microbiological analyses [44]. Ethical approval was granted from the Animal Experimentation Ethical Review Committee (AEERC), Faculty of Biological Sciences, University of Dhaka under the Reference No. 71/Biol.Scs./2018-2019 (approved on 14 November 2018).

### 2.2. Isolation and Identification of Pathogenic E. coli

We adopted the method of Knobl et al. [45] and modified protocol of Food and Drug Administration Bacteriological Analytical Manual (FDA-BAM) guidelines for isolating and identifying APEC [46]. In brief, one loopful inoculum from each sample was inoculated into previously prepared nutrient broth (NB) composed of yeast extract (2 gm/L), peptone (5 gm/L), and sodium chloride (5 gm/L), and incubated for 24 h at 37 °C. A small amount of inoculum from NB was streaked onto MacConkey (MC) agar, and Eosin Methylene Blue (EMB) agar for 24 h at 37 °C for selective growth. The colonies showing characteristic metallic sheens on the EMB agar (3–5 colonies from each sample) were selected to further biochemical tests (indole, methyl-red, catalase, citrate, and Voges–Proskauer) for confirmatory identification of *E. coli* [44]. 

### 2.3. Phenotypic Virulence Assays

The assays for virulence properties of *E. coli* were performed according to the FDA-BAM guidelines.

#### 2.3.1. Congo Red Binding Assay

The Congo red binding (CRB) ability of the APEC isolates was determined using agar plates supplemented with 0.003% CR dye (Sigma, St. Louis, MO, USA), and 0.15% bile salt. Bacterial suspension (5 µL) was streaked onto the plates, and the plates were incubated at 37 °C for 24 h. The strong biofilm-producing isolates were visualized as deep brick red-colored. For more confirmation, colonies were also consecutively examined following 48 and 72 h of incubation, and the results were interpreted as +++, ++ and + depending on their color intensity [23,32]. 

#### 2.3.2. Production of Hemolysins and Swimming Motility Assays

The *E. coli* hemolytic activity of the CRB positive isolates was evaluated by streaking blood agar base plates supplemented with 5% sheep blood. After 24 h incubation at 37 °C, plates were examined for signs of β-hemolysis (clear zones around colonies), α-hemolysis (a green-hued zone around colonies) or γ-hemolysis (no halo around colonies) [47]. We performed motility assay by following the previously described protocols [48]. In brief, bacterial cultures were stabbed onto motility indole urease (MIU) soft agar motility tubes (0.5% agar), and the tubes were incubated for 24–48 h at 37 °C. Finally, we measured the bacterial motility haloes after 24 and 48 h of incubation [48].

#### 2.3.3. Biofilm Formation Assay

The biofilm formation (BF) assay of randomly selected 82 CRB positive *E. coli* isolates was performed by using 96-well microtiter plate methods to quantitatively measure the attachment and BF on solid surfaces (plastic) during static conditions [49]. The assay was performed in duplicate in the 96-well tissue culture plates. The observed optical density (OD) was evaluated to determine the BF ability of the isolates on a 4-grade scale (non-adherent, weakly adherent, moderately adherent, and strongly adherent). These 4 grades were determined by comparing OD with cut-off OD (ODc) (three standard deviation values above the mean optical density of the negative control). Furthermore, the biofilm surface after 24 h was stained with biofilm viability kit to observe the proportion of live or active cells (fluorescent green), and dead or inactive cells (fluorescent red) using fluorescence microscope (at general magnification of 40× objective), and DP73 digital camera (40× objective). After that the microscopic images were analyzed by the ImageJ software bundled with Java (version: 1.8.0_112) [49].

### 2.4. DNA Extraction

Genomic DNA of *E. coli* was extracted from overnight culture by the boiled DNA extraction method [50]. Briefly, the samples were centrifuged at 15,000× *g* for 15 min [50]. We eliminated the supernatant, resuspended the pellets in filtered (22 microliter) distilled water, and centrifuged at 15,000× *g* for 10 min. Again, the supernatant was eliminated, the pellets were resuspended in 200 µL of filtered distilled water, subjected to boiling at 100 °C in a heat block for 10 min, cooled on ice for 10 min, and centrifuged at 15,000× *g* for 10 s [51]. The extracted DNA (120–150 microliter per sample) was quantified using a NanoDrop ND-2000 spectrophotometer.

### 2.5. Molecular Typing Methods

Species identification of *E. coli* in 174 CRB-positive isolates were confirmed using *E. coli* specific PCR, such as random amplification of polymorphic DNA (RAPD) [6], box elements (BOX-PCR) [8] and enterobacterial repetitive intergenic consensus PCR (ERIC-PCR) [10]. The optimized protocol for RAPD PCR was performed by using (5′-GCGATCCCCA-3′) primer [51], while the ERIC-PCR was done with ERIC1R (5′-ATGTAAGCTCCTGGGGATTCAC-3′) and ERIC2 (5′-AAGTAAGTGACTGGGGTGAGCG-3′) primers [10], and the BOXA1R (5′-CTACGGCAAGGCGACGCTGACG-3′) primer was used for BOX-PCR [8].

### 2.6. Clermont’s Phylogenetic Typing 

The isolates were subjected to phylotyping using the Clermont et al. method [9]. The reaction mixture (15 μL) contained about 7.5 μL of GoTaq^®^ G2 Green Master Mix (Promega, Madison, WI, USA), 1.0 μL of primer, 4.5 μL nuclease-free water (Promega, USA), and 1.0 μL of template genomic DNA. Cycling conditions were as follows: A total of 5 min initial denaturation at 94 °C, followed by 30 cycles of 5 s at 94 °C, 25 s at 59 °C; with a final extension of 5 min at 72 °C. PCR products were separated and visualized on 1.2% agarose gels in 1-Trisborate- EDTA (TBE) buffer using ethidium bromide staining. Following electrophoresis, the gel was photographed under UV light, and the strains were assigned to the phylotypes B2 (*chu*A+, *yja*A+, *arp*A-), D (*chu*A+, a*rp*A*+, yja*A-), B1 (*chu*A-, *arp*A+, TspE4.C2+), C (*chu*A-, *yja*A+, *arp*A+, TspE4.C2-), E (*chu*A+, *arp*A+, *yja*A-, TspE4.C2+), F (*chu*A+, *yja*A-, *arp*A-, TspE4.C2-), or A (*chu*A-, *yja*A+/-, *arp*A+, TspE4.C2-). For better strain-level discrimination, the subgroups or phylotypes were determined as follows: Subgroup A0 (group A), *chu*A-, *yja*A-, *arp*A+, TspE4.C2-; subgroup A1 (group A), *chu*A-, *arp*A+, *yja*A+ TspE4.C2-; group B1, c*hu*A-, *arp*A+ *yja*A-, TspE4.C2+; subgroup B2_2_ (group B2), *chu*A+, *arp*A-, *yja*A+, TspE4.C2-; subgroup B2_3_ (group B2), *chu*A+, *arp*A -, *yja*A+, TspE4.C2+; subgroup D1 (group D), *chu*A+, *arp*A+, *yja*A-, TspE4.C2- and subgroup D2 (group D), *chu*A+, *yja*A-, *arp*A +,TspE4.C2+ [52].

### 2.7. Ribosomal Gene (16S rRNA) Sequencing and Phylogenetic Analysis

*E. coli* isolates from each phylotype along with other typing methods (RAPD, ERIC, and BOX-PCR) were selected for 16S rRNA gene PCR using universal primers (27F, 5’-AGAGTTTGATCMTGGCTCAG-3’ and 1492R, 5’-TACGGYTACCTTGTTACGACTT-3′), followed by sequencing at First Base Laboratories SdnBhd (Malaysia) using Applied Biosystems highest capacity-based genetic analyzer (ABI PRISM^®^ 377 DNA Sequencer) platforms with the BigDye^®^ Terminator v3.1 cycle sequencing kit chemistry. Initial quality control of the generated raw sequences was performed using SeqMan software (version 6), and aligned with 13 relevant reference sequences retrieved from NCBI database using Molecular Evolutionary Genetics Analysis (MEGA) version 7.0 for bigger datasets [53]. Evolutionary distances were computed using the Kimura–Nei method, and the phylogenetic tree was constructed by applying the neighbor-joining method [54]. The percentage of replicate trees in which the associated taxa clustered together in the bootstrap test (1000 replicates) is shown next to the branches. 

### 2.8. Antimicrobial Sensitivity Testing

Kirby-Bauer disc diffusion method on Mueller-Hinton agar was used for antimicrobial susceptibility testing of various *E. coli* pathotypes according to the guidelines of the Clinical and Laboratory Standards Institute [55]. Antibiotics were selected for susceptibility testing corresponding to a panel of antimicrobial agents of interest to the poultry industry and public health in Bangladesh. Randomly selected representative 114 APEC isolates from all phylotypes were evaluated for antimicrobial susceptibility to 13 antibiotic discs, which belonged to 11 different antibiotic classes including penicillins (ampicillin, 10 µg); tetracyclines (doxycycline, 30 µg; tetracycline, 30 µg); nitrofurans (nitrofurantoin, 300 µg); lipopeptides (polymexin B, 30 µg); monobactams (aztreonam, 30 µg); quinolones (ciprofloxacin, 10 µg; nalidixic acid, 30 µg); cephalosporins (cefoxitin, 30 µg), penems (imipenem, 10 µg); aminoglycosides (gentamycin, 10 µg); phenols (chloramphenicol, 30 µg); sulfonamide (sulfonamide, 250 µg), and macrolides (azithromycin, 15 µg) (Oxoid, UK). Finally, the findings were recorded as susceptible, intermediate, and resistant according to Clinical and Laboratory Standards Institute [55] break points.

### 2.9. Detection of Virulence Genes

We surveyed 13 virulence genes (VGs) that are usually studied in APEC strains. The selected genes included: *eae* (Intimin) [29], *stx1* (shiga toxins 1) [29], *stx2* (shiga toxins 2) [29], *fim*H (type 1 fimbriae) [32], *hly*A (hemolysin) [32], *pap*C (P fimbriae) [32], *lt* (heat-labile enterotoxin) [56], *bfp*A (bundle-forming pilus) [29], *crl* (curli fimbriae) [23,31,45], *uid*A (β-d-glucuronidase) [29,32] *agg*R (aggregative adherence regulator) [57], *ial* (invasion-associated locus) [29], and *cjr*C (putative siderophore receptor) [29,32]. Each PCR reaction contained 2 μL DNA template (300 ng/μL), 10 μL PCR master mix 2X (Go Taq Colorless Master Mix), and 1 μL (100 pmol/μL) of each primer in each tube. The PCR amplifications were conducted in thermocycler, and the cycling conditions were identical for all the samples as follows: Temperature at 94 °C for 5 min; 35 cycles of 1 min at 94 °C, 1 min at 50–60 °C, and 1 min at 72 °C; and 72 °C for 7 min. PCR amplicons were visualized on 1.5% agarose gel prepared in 1× TAE buffer. After gel electrophoresis, the images were captured using Image ChemiDoc™ Imaging System (Bio-Rad, Hercules, CA, USA) [29,32].

### 2.10. Plasmid DNA Isolation

The plasmids DNA of *E. coli* isolates were isolated following the method previously reported by different researchers [58]. Briefly, plasmid DNA of 45 randomly selected *E. coli* isolates was extracted using Wizard^®^ Plus SV Mini preps plasmid DNA Purification kit (Promega, USA) according to manufacturer’s instruction, and was electrophoresed in 0.8% agarose gel [58]. *E. coli* V517 strain was used in the present study for molecular weight determination as control [59].

### 2.11. Discriminatory Index (D)

The discriminatory index (DI) was measured for each sample by entering data onto the following site: http://insilico.ehu.es/mini_tools/discriminatory_power/index.php [60]. 

### 2.12. Correspondence Analysis

Correspondence analysis (CA) was used to study and compare the categories of molecular typing (phylotypes), the origin of sample groups (poultry feed and water, handler, egg surface, droppings, cloacal swab, liver), and pathogenic intensities (pathogenic genes, BF and drug sensitivity) using Pearson correlation tests through the IBM SPSS Statistics 20.0 package. A two-dimensional graph was used to show the relationship between the categories in CA analysis, where the value of the third dimension is shown in parenthesis [61]. The association of different sample types with molecular typing and pathogenic intensities were represented using the circular plot. The plot was visualized using OmicCircos [49].

### 2.13. Statistical Analysis

We used the SPSS software for Windows, version 20.0 (SPSS Inc., Chicago, IL, USA) for statistical analysis [62]. Comparison among frequencies of occurrence of each phenotypic or genotypic feature in *E. coli* isolates from poultry originated samples were carried out by contingency table χ2 tests (at *p* < 0.05). To analyze motility, hemolysin, and biofilm assay data, we applied one-way analysis of variance (ANOVA), and two-way ANOVA was performed to analyze the typing methods and antimicrobial susceptibility tests. The association between molecular typing and sample types, and molecular typing and pathogenic intensities were calculated using the χ2 tests. The result was considered to be significant at *p* ≤ 0.05. 

## 3. Results

### 3.1. Phenotypic Characterization of APEC: Isolation and Identification

A total of 130 poultry samples (droppings, 30; cloacal swabs, 27; feeds, 11; handler’s swab, nine; egg surface swab, 10; feeding water, 10; liver, 33) were screened for phenotypic identification of *E. coli*. According to the microbiological analysis, 392 isolates were obtained through selective identification in EMB and MacConkey agar (metallic sheen on EMB agar plates, and pink colonies on MacConkey agar), and biochemical tests followed by Congo red binding (CRB) assay. Results from the CRB assay revealed 174 isolates as avian-pathogenic *E. coli* (APEC), of which 43 and 131 isolates belonged to healthy and diseased birds, respectively. Of these APEC isolates, 46.55%, 39.38%, and 32.69% were from Narayangonj, Manikgonj, and Narsigdi districts, respectively. The distribution of pathogenic *E. coli* in different samples have been represented in Figure 1. In this study, the droppings from birds had highest (33.33%) amount of APEC isolates followed by liver (19.54%), cloacal swab (17.82%), handler’s swab (10.34%), feeding water (9.20%), feeds (5.17%), and egg surface swabs (4.60%) (Figure 1; Appendix A). The frequency of the detected bacterial isolates also significantly (*p* = 0.019) varied within the three sampling sites.

### 3.2. Genetic Diversity of APEC by Molecular Fingerprinting

In this study, we used RAPD, ERIC, and BOX-PCR to analyze the genetic diversity and relatedness in 174 isolates of APEC originated from different poultry samples. RAPD showed 10 different patterns among the isolates, and the reproducibility of the RAPD technique was analyzed by repeated testing (Appendix A). In case of ERIC and BOX-PCR assays, the number of DNA bands for different APEC isolates were one to five and one to seven, respectively, and thus the isolates were differentiated into eight and nine groups through ERIC and BOX-PCR, respectively (Appendix A). Therefore, RAPD fingerprint of DNA showed the highest genetic diversity among the isolates followed by BOX and ERIC PCR. The discriminatory indices (DI) for RAPD, ERIC and BOX-PCR were 0.8707, 0.8371, and 0.8591, respectively, for all isolates. The diversity of the isolates, measured through the principle component analysis (PCA) revealed that the genetic diversity of the APEC isolates did not vary significantly (*p* > 0.05) according to molecular typing systems, since in all typing methods group 1–4 clustered in the same quadrant of the PCA plot (Figure 2).

### 3.3. Phylogenetic Distribution of APEC in Poultry Isolates

The phylogenetic investigation of 174 APEC isolates identified five phylotypes (A1, B1, B2_2_, B2_3_, and D2). However, any gene combinations for phylotype E, C, and F were absent in the APEC isolates of the current study (Appendix A, Appendix A). In the comparative analysis, we found that majority of APEC isolates were affiliated to phylotype B2_3_ comprising 37.36% (65/174), followed by A1 (33.91%), D2 (11.49%), B2_2_ (9.20%) and B1 (8.05%) (Figure 3, Appendix A). In this study, phylotype A1 was the more prevalent (55.81%) in the samples from healthy birds, while phylotype B2_3_ (41.22%) had the higher prevalence in the samples of infected birds, followed by A1 (26.72%), D2 (14.50%), B2_2_ (11.45%), and B1 (6.11%) (Appendix A). Our results demonstrated that APEC phylotypes distribution differed significantly (*p* = 0.002) across the study areas. The isolates from Manikgonj district were segregated into phylotype B2_3_ (36.84%), A1 (30.26%), B1 (14.47%), D2 (11.84%), and B2_2_ (6.58%), while those from Narayangonj district were segregated into phylotypes A1 and B2_3_ (37.03%, each), B22 (12.35%), D2 (9.99%), and B1 (3.70%). Conversely, none of the isolate from Narsingdi district harbored phylotype B1. In this study, the phylotype B2 (B2_2_ and B2_3_) and A1 were more prevalent in all of the APEC isolates; however, B1 and D2 phylotypes were not found among the isolates of poultry feed and egg surface swab (Appendix A). 

### 3.4. Biofilm Formation (BF) Assay

As quantified in crystal violet assay, biofilm-forming bacteria were divided into four groups based upon OD_600_ of the bacterial biofilm: non-biofilm forming (NBF), weak biofilm forming (WBF), moderate biofilm forming (MBF), and strong biofilm forming (SBF) bacteria. In the current study, the average OD of the negative control was 0.028 ± 0.002, and the cutoff OD value was set as 0.045. The isolates having OD value ≤ 0.045 were considered as NBF. We found that 81.71% of the APEC isolates were biofilm formers representing 16.42% and 83.58% isolates from healthy and diseased birds, respectively (Appendix A). By comparing the category of BF, we demonstrated that 30.49%, 26.83%, 24.39%, and 18.29% isolates were SBF, MBF, WBF, and NBF, respectively (Figure 4A, Appendix A). Of the SBF isolates, 16.0% and 84.0% were found in healthy and infected birds, respectively (Appendix A). Microscopic observation followed by 3D image analysis revealed that the intensity of green fluorescence remained higher indicating that a large number of cells were viable and attached to the surface (Figure 4B). The isolates having SBF properties belonged to APEC (B2_2_, 42.86; B2_3_, 29.73; D2, 54.55), and thus, D2 isolates of APEC had the highest (54.55%) SBF ability (Figure 4A). Notably, all of the isolates from phylotype D2 had BF ability followed by phylotype B2 (86.36%). Interestingly, in both healthy and diseased birds, phylotype B2_3_ showed the highest BF ability (Appendix A).

### 3.5. Distribution of Virulence Genes of APEC in Poultry Isolates

The possible association of different virulent genes (VGs) was screened through PCR in 123 APEC isolates according to their phylogroups. In this study, thirteen probable APEC associated VGs including the diarrheagenic and septicemic genes encoding for *eae*, *stx*1, *stx*2, *fim*H, *hly*A, *pap*C, *lt*, *bfp*A, *crl*, *uid*A, *agg*R, *ial*, and *cjr*C were screened. The virulence genotyping showed that none of the APEC isolate harbored genes coding for *eae*, *stx*1, *stx*2, *hlyA*, *bfp*A, and *fli*C (Appendix A). The distribution of the six apparently higher prevalent VGs, such as *crl*, *fim*H, *ial*, *pap*C, and *cjr*C, is shown in Figure 5, and Appendix A. Among the identified VGs, *uid*A was present in all of the APEC phylotypes (100%) while *crl*, *fim*H, and *ial* were found in 80 to 100% of the isolates examined. The abundances of *crl*, *fim*H, and *ial* were 100.0% (for each) in the isolates of phylotype D2 of APEC while the prevalence of *pap*C and *cjr*C genes among these D2 phylotypes was 77.78% and 77.22%, respectively (Figure 5). On the other hand, the prevalence of *pap*C gene was 50.0%, 39.13%, and 26.68%, respectively, in B2_2_, B2_3_ and A1 phylotypes. The *cjr*C gene was found in 41.67%, 41.30%, and 13.16% isolates of the phylotype B2_2_, B2_3_, and A1, respectively (Figure 5). However, none of the isolates of phylotype B1 possessed these two (*pap*C and *cjr*C) genes. Thus, our present results revealed significant (*p* < 0.05) association between two VGs (*pap*C and *cjr*C), and pathogenic phylotypes (D2, B2). Conversely, only two phylotypes (phylotype A1 and phylotype B2_3_) from healthy birds harbored only two VGs (*pap*C and *cjr*C) (Appendix A). Therefore, our study showed that VGs *pap*C and *cjr*C, on average, were less abundant in all phylotypes with the exception of phylotype D2 (~78%) (Figure 5). 

### 3.6. Genetic Diversity of APEC Isolates Based on Ribosomal Gene (16S rRNA) Sequencing

Nucleotide sequences obtained from 11 APEC isolates according to phylotyping and molecular typing (RAPD, ERIC, and BOX-PCR) along with 13 previously reported reference sequences retrieved from the NCBI database were used to generate a phylogenetic tree. The results obtained from the phylogenetic analysis exhibited two different clades based on RAPD grouping, and we referred to them as clade A and clade B, which contained five and six isolates of APEC, respectively (Figure 6). The strains of APEC found in clade A had 98–100% similarity with *E. coli* APEC O78, and *E. coli* EHEC strains; whereas the strains found in clade B had closest relationship with *E. coli* O169:H41 strain. Interestingly, identified strains of the APEC from B2 and D2 phylotypes were found in both clades, and the strains from phylogroups A1 and B1 were grouped into clade A only (Figure 6).

### 3.7. Antibiogram Profiling of Circulating APEC Phylotypes

Antibiotic susceptibilities of randomly selected 114 isolates of APEC were carried out using 13 different antibiotics belonging to 11 groups (Table 1). The tested isolates showed a varying degree of resistance towards these antibiotics. The antibiogram profiling of the current study revealed that most of the APEC isolates (78.95%) were resistant to doxycycline, followed by nalidixic acid (76.32%), tetracycline (75.44%), ampicillin (74.76%), nitrofurantoin (63.16%), chloramphenicol (51.75%), cefoxitin (41.23%), ciprofloxacin (44.75%), sulfonamide (44.74%), azithromycin (31.58%), gentamycin (26.32%), imipenem (22.81%), and polymyxin B (7.78%). In regards to phylotyping, 61.96% isolates in phylotype B1 were resistant to at least three tested antibiotics, while, 55.24%, 53.85%, 51.16%, and 45.58% of the isolates of phylotypes D2, B2_2_, B2_3_, and A1, respectively, were found to be resistant against ≥3 tested antibacterial agents (Table 1). However, in the comparative analysis, we found that 85.0% isolates belonging to phylotype A1 were resistant to tetracycline, and 77.5%, 70.0%, and 60.0% isolates of this phylotype were resistant to doxycycline, ampicillin, and nalidixic acid, respectively. Resistance to tetracycline, doxycycline, nalidixic acid, and ampicillin was 83.72%, 79.07%, 76.74%, and 74.42% in isolates from phylogroup B2_3_, and 81.82%, 72.72%, 72.72%, and 90.91% in isolates of phylotype B2_2_, respectively (Table 1). The resistance tendency of the phylotype B1 to ampicillin, doxycycline, ciprofloxacin, and gentamicin was higher than the resistance rate of the phylotype D2. However, this B1 phylotype had a lower resistance to tetracycline, nitrofurantoin, nalidixic acid, and polymyxin B. Thus, on an average, the isolates of B1, B2_2_, and D2 phylotypes were more resistant than those of phylotypes B2_3_ and A1 (Table 1). However, we did not find any significant differences (*p* > 0.05) in MDR properties among the isolates of healthy and infected birds, which indicated that indiscriminate and unauthorized use of antibiotics in the poultry farms of the Bangladesh. Furthermore, no significant differences were observed in the prevalence of VGs between nitrofurantoin-resistant and susceptible APEC strains except *pap*C, which remained more prevalent in the nitrofurantoin-susceptible strains (38.24%) than the nitrofurantoin-resistant strains (24%) (Appendix A).

### 3.8. Existence of Plasmid

Plasmid profiling of 45 randomly selected APEC isolates based on their genetic diversity, and AMR properties (Appendix A), showed that 73.33% of the APEC isolates were plasmid bearing. Among these plasmid-harboring isolates, 9.09% and 90.91% isolates belonged to healthy and diseased chicken’s samples, respectively (Figure 7, Appendix A). The molecular weight of these plasmids varied from >30 to 2.1 kb. All of the plasmid-harboring isolates showed multiple plasmid bands with size of 3 to 7.3 kb. However, the common size of plasmids was 3.9 to 5.6 kb, as detected in all of the plasmid-bearing strains (Figure 7). However, we demonstrated weak correlation (*p* = 0.07) between plasmid-bearing APEC isolates with their phylogroups. With co-existence of the plasmid-bearing genes and phylotypes, our results showed that 100% of isolates belonging to phylotype D2 of APEC harbored plasmids of variable bands size and weight. Conversely, 87.5%, 83.33%, 60%, and 50% of plasmid-possessing isolates belonged to phylotypes B2_2_, B2_3_, B1, and A1 of APEC, respectively (Figure 7, Appendix A).

### 3.9. Associations between Sample Types, APEC Phylotypes, and Pathogenic Intensity

The relationship analysis of the strength of sample types, molecular typing, and pathogenic intensity was performed to determine possible associations among the isolates. In terms of molecular typing methods, we demonstrated that droppings (*p* = 0.02), cloacal swabs (*p* = 0.037), and internal organ liver (*p* = 0.041) had significantly stronger correlation with RAPD followed by BOX (*p* = 0.036, *p* = 0.043, *p* = 0.047, respectively), phylotypes (*p* = 0.045, *p* = 0.048, *p* = 0.033, respectively), and ERIC (*p* = 0.13, *p* = 0.017, *p* = 0.29, respectively) (Figure 8). While analyzing the pathogenicity and antibiogram profile of the tested isolates, sample categories had significantly higher correlation with CRB (*p* = 0.008) followed by biofilm formation (*p* = 0.02), drug sensitivity (*p* = 0.03), and virulence genes (*p* = 0.06). We also used correspondence analysis (CA) to measure the degree of relationship between the categories of the phylotypes, origin of samples (poultry feed and water, handler, egg surface, droppings, cloacal swab, liver), and pathogenic intensity (drug sensitivity). The bi-dimensional representation of phylotypes distribution in each of the seven sample categories is shown in Figure 9. The bi-dimensional representation explains 100% of the total variation, with 68.55% explained by first dimension, and 31.45% by the second dimension. In contrast, the CA for the drug sensitivity and biofilm formation representation explains 87.25% variation by the first dimension, and 12.75% by the second dimension. Moreover, in the current study, correspondence analyses also demonstrated significant correlation among all of the phylotypes with Congo red binding (*p* = 0.008), antimicrobial resistance (*p* = 0.03), biofilm-formation (*p* = 0.02), and virulent genes (*p* = 0.06).

## 4. Discussion

The avian colibacillosis, caused by different strains of *E. coli*, is considered as one of the major threats to the poultry industry and public health worldwide [63]. The devastating impacts of colibacillosis are particularly evident in the poultry farms of developing countries because of the poor hygienic practice and management [64]. In this study, we tested avian pathogenic *E. coli* (APEC) isolates from different poultry samples with regard to their phylotypes, phenotypic and genotypic virulence traits, and antimicrobial resistance. The findings of the current study provided evidence that the poultry farms could indeed be contaminated with multidrug-resistant (MDR) APEC phylotypes, especially with the potentially pathogenic B2 and D2 phylotypes. This is particularly alarming for Bangladesh because of having a high disease burden, emergence of resistance traits, and the confluence of prevailing socio-economic, demographic, and environmental factors [40,41]. This is the first study to report the association of multidrug-resistant APEC phylotypes in avian colibacillosis in different poultry farms of Bangladesh. A likely explanation to this high level of MDR potential in pathogenic APEC isolates could be the improper disinfection management, lack of empty period of implementation between flocks, lack of knowledge about cleanliness, impure poultry feed and feeding environment, use of contaminated water, and extensive use of antimicrobials in chickens, often without veterinary prescription, as reported in many earlier studies [40,41,42,43]. In the present study, 392 probable APEC isolates were obtained from 130 poultry samples (droppings, cloacal, feed, handler, egg, water, liver) collected from three districts (Narsingdi, Narayangonj and Manikgonj) of Bangladesh, with 44.39% prevalence of colibacillosis. In Bangladesh, several earlier investigations in broiler chickens with colibacillosis reported 20% to 80% prevalence of this infectious disease [36,40,41,43,65]. The prevalence of colibacillosis in other countries like Nepal, China, Brazil, and India ranged from 30% to 80% [4,66,67,68]. The predisposing epidemiologic factors such as geographic locations, farm housing types, varying sample collection, transportation and preservation methods, and management practices are likely to contribute to the differences in frequency of pathogenic APEC isolation [67].

### 4.1. Circulatory Molecular Phylotypes of the APEC in Bangladesh

Phylogenetic analysis revealed that most of the APEC isolates were from phylotype B2_3_ followed by A1, D2, B2_2_, and B1, which provided a credible reference on the ecological distribution and genetic evolution of different pathogenic strains in the poultry farms of Bangladesh. Considering the elevated rate of APEC B2 and A1 phylotypes detected in this study, and consistent with previous reports [3,31,42,52], we may infer that poultry samples could be a potential reservoir of APEC. Another study in Sri Lanka on phylogenetic diversity of APEC isolates from septicemic broiler and layer cases reported that the APEC isolates belonged to A (71.00%), B1 (4.10%), B2 (7.90%), and D (18.70%) phylogroups [16]. However, most of the recent studies reported phylotypes A and D as the most abundant APEC phylotypes isolated from poultry, such as in Italy [12], China [15], Canada [13], and Iran [14], indicating that the frequency of phylotypes might vary among different geographic regions. Globally, phylotypes B2 and D are classified as pathogenic *E. coli* [9], and our present findings are in accordance to these reports. Although recent studies that utilized the updated method by Clermont et al. [9] are scarce, Logue et al. [42] classified APEC isolates according to the new phylogenetic typing, and concluded that strains in A and B1 group had a lower pathogenic potential. The lower prevalence of phylogenetic group D2 was also reported earlier [12,14]. In addition to phylotyping, three molecular typing methods, such as RAPD, ERIC and BOX-PCR, were used to reveal the genetic relatedness among the APEC isolates [10]. Though the MLST technique is one of the important and widely used tool for APEC characterization globally, we did not utilize this technique in APEC phylotyping in consideration of some of its inherent disadvantages [11,27]. 

### 4.2. Pathogenic Properties of the Circulating APEC

APEC isolates often carried a broad range of virulence genes (VGs) that may enable their pathogenicity in avian colibacillosis. These include production of adhesions, toxins, siderophores, iron transport systems, and invasins [26,31,32]. Several VGs such as *ial*, *pap*C, *fim*H, *crl* are important in APEC adherence [32]. In this study, many of the APEC isolates belonging to A1, B1, B2, and D2 phylotypes carried one or more virulence genes, which were represented by *pap*C, *cjr*C, *crl*, *ial*, *fim*H and *uid*A. The phylogroup B2 and D2 (pathogenic strains) harbored all of these virulence determinants, while phylotype A and B1 (usually found in commensal strains) possessed few VGs. Smith et al. [69] reported that the phylotype B2 and D possessed several pathogenicity-associated islands and express multiple virulence factors, such as adherence factors including biofilm production, supporting our current investigation. In the present investigation, we revealed that the distribution of six VGs was different, and a large proportion of adherence genes had strong biofilm-formation ability. Therefore, a positive correlation among these genes, biofilm phenotypes, and phylotypes was observed. In the current study, the VGs *uid*A, *crl*, *fim*H, and *ial* were found in 80% to 100% of the APEC isolates examined. These results are in accordance with many of the previously published studies [23,26,27,31]. Moreover, the presence of similar VGs in APEC isolates proposed that APEC isolates can act as zoonotic pathogens, and reservoirs of virulence causing human infections [23,32,36]. The *fim*H virulence factor is seemed to be an essential unit for protecting the APEC isolates against host immune system but the exact role of *fim*H in the pathogenicity of APEC isolates remains debatable with incompatible results [70]. The CRB assay has been used extensively to distinguish curli-producing bacteria from non-curliated bacteria [23,31]. Furthermore, curli fibers are protease resistant, and bind to CR and other amyloid dyes [23]. The curli-negative mutant isolates had less adherence colonization, invasion, and persistence to chicken tissues, recommending curli as a virulence factor [23,26,27,32]. Therefore, it can be supposed that most APEC isolates are curliated [23,27]. The relative abundance of the *cjr*C gene seemed to be positively correlated with the AMR profile of the isolates of phylotypes D2, B2_3_, and A1. In another study, Zhao et al. [71] found that the prevalence of *iro*N gene (Salmochelins related) in Cefoxitin-susceptible UPEC isolates was significantly higher than the resistant genes, and nitrofurantoin-resistant isolates had reduced VGs compared with susceptible strains. The *pap*C genes; however, had relatively lower abundance among the APEC isolates, and we did not reveal any differences in the prevalence of VGs between nitrofurantoin-resistant and susceptible APEC strains. These results are in line with the findings of many other studies [32,71]. Furthermore, most of the virulence determinants identified in this study could be acquired by horizontal transmission without disrupting the clonal lineage. Nevertheless, it is yet unknown whether the acquisition of these genes is associated with the pathogenesis of a particular microbe or if a specific genetic background is required for the transfer and expression of these genes [16]. Moreover, BF is an important virulence factor for APEC strains, and contributes to the resistance to different classes of antimicrobials [49,72]. APEC strains identified in this study showed broad spectrum of antimicrobial resistance, and possessed biofilm-forming abilities, which might be the potential factors for colibacillosis in the poultry farm, persistence of the disease, and increased risk of transmission to non-infected birds. However, pathogenic potentials of the APEC-associated VGs have not been demonstrated using in vivo animal trials, which is one of the drawbacks of the current study.

### 4.3. Correlations between Phylotypes and MDR of Circulating APEC

Colibacillosis in the poultry farms might be prevented and/or controlled by the rational therapeutic use of antimicrobials. However, evolution of MDR APEC strains along with the transmission of resistance genes has created challenges in reducing the risk of APEC infections [73]. Regarding antimicrobial resistance exhibited by the isolates of different phylotypes, our results indicated that phylotype A1 isolates were more susceptible than the isolates of other phylotypes. Conversely, the isolates of phylotype B1 displayed the highest antimicrobial resistance pattern. Previous studies reported that although being more virulent, the isolates of phylotype B2 were more susceptible to antibiotics [4]. However, we found that resistance to doxycycline, ampicillin, and nalidixic acid was common in group B2 isolates, while the phylotype A1 remained resistant to tetracycline only. Therefore, our present findings demonstrated that strains belonging to phylotypes B1, D2, and B2_2_ were carrying more resistance and/or virulent properties than the strains of phylotype B2_3_ and A1, corroborating the findings of Iranpour et al. [52] and Moreno et al. [74]. In this study, we demonstrated that all of the APEC phylotypes possessed MDR properties, which did not comply with the previous findings of Etebarzadeh et al. [52] and Iranpour et al. [75], who reported that only the phylotype B2 of APEC isolates could bear MDR phenomena. This variation could be explained by the horizontal transfer of resistance genes through plasmids across the APEC strains. These findings; therefore, imply that in the poultry industry of Bangladesh a large number of poultry samples might act as a reservoir for such resistant strains. Unfortunately, we did not find any single APEC isolate showing sensitivity to all of the 13 antibiotics tested, which might be due to the widespread, indiscriminate, and long-term use of similar drugs in the poultry farms [73,76]. Nonetheless, all of the (100%) plasmid-bearing strains found in this study were multi-drug resistant. Even though, isolates that did not bear any plasmid DNA (26.67%) were also resistant to at least three or more antimicrobials tested [77], which could make it more possible for a susceptible bacterium to acquire resistance factors through conjugation or transformation [73,76]. However, the number of plasmids found in a particular isolate would not necessarily indicate the level of MDR properties of the isolate, which might also be one of the predisposing causes of spreading and developing MDR properties among poultry population in Bangladesh. 

### 4.4. Correlations among Sample Types, Molecular Typing Methods, Pathogenic Intensity, and MDR Properties of Circulating APEC

The phylotype distribution of circulating APEC isolates was influenced by many factors, such as sample types, CRB, biofilm formation (BF), and VGs [36]. The correspondence analysis (CA) and circular visualization revealed stronger association between pathogenic intensity (CRB, BF, VGs) and molecular typing (phylotyping), and these phenomena of APEC isolates might be associated to MDR properties of this bacterium in the poultry farms of Bangladesh, as supported by previous studies [48,61,78]. Our results indicated that phylotype B2 was the main circulating APEC followed by phylotype A1 (Figure 8 and Figure 9, Appendix A). However, both the results of CA and circular plot also showed that APEC phylotypes B2, B1, D2, and A1 were predominantly isolated from droppings and cloacal samples. The CA analysis did not display stronger association between B2 and D2 APEC phylotypes, their BF ability, and VGs; however, further visualization using the circular plot showed that these two phylotypes had higher potentials for BF, and harbored more VGs than other APEC phylotypes identified in this study (Figure 8 and Figure 9, Appendix A). These findings also corroborated with many earlier studies [48,78]. Furthermore, both CA and circular plot visualization showed that all of the APEC phylotypes isolated from different poultry samples possessed MDR phenomena, as also reported in many recent studies [73,76]. Therefore, high prevalence of antibiotic-resistant APEC strains, and their associations with sample types, molecular typing methods, pathogenic intensity and MDR properties, suggest an alternative approach of organic antimicrobial compounds, and/or metals usages, and the rotational selection and judicious use of antibiotics in the poultry farms in Bangladesh.

## 5. Conclusions

The identification of VGs and AMR from different samples of avian colibacillosis reveals the great importance of APEC zoonotic potential. Our results showed that five phylogroups were prevailing among the APEC isolates, and of them, phylotypes A1 and B2 were the most common groups. Phylogenetic analysis revealed two distinct clades (clade A and clade B) of APEC. Phylogroups B2 and D2 were found in strains of both clades, and phylogroups A1 and B1 were found in clade A only. Antibiogram profiling revealed that 100% of the isolates, and the majority of the phylotypes (>50) were resistant to at least three antibiotics. Our data demonstrated that relatively high MDR levels among all of the APEC phenotypes is a serious concern in Bangladesh, and may lead to several adverse effects on animals, humans, and the environment. Furthermore, the APEC isolates and their associated phylogroups in the present study harbored a set of VGs in a relatively higher proportion along with potential BF ability. These two phenomena of the APEC phylotypes could be associated with their MDR properties, which could cause a serious public health problem. However, future research should be done to better understand the flow of antimicrobials usage, monitoring the spread of ARGs, and VGs using larger population size in different ecosystems of the poultry sectors of Bangladesh.

## Figures and Tables

**Figure 1 microorganisms-08-01135-f001:**
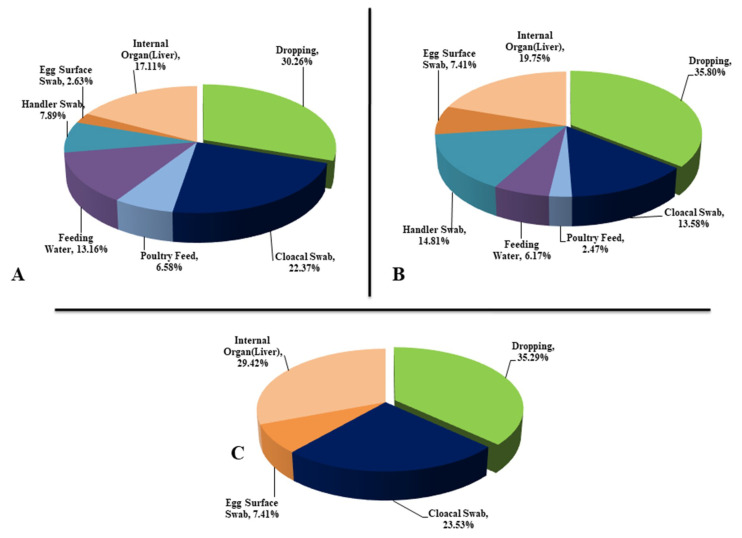
Prevalence of avian pathogenic *Escherichia coli* (APEC) in different types of poultry samples. Microorganisms identified in different types of avian samples belonged to different locations: (**A**) (Dhamrai, Manikgonj); (**B**) (Rupganj, Narayangonj); and (**C**) (Monohardi, Narshingdi). The Dhamrai, Rupganj and Monohardi regions included 76, 81, and 17 isolates, respectively.

**Figure 2 microorganisms-08-01135-f002:**
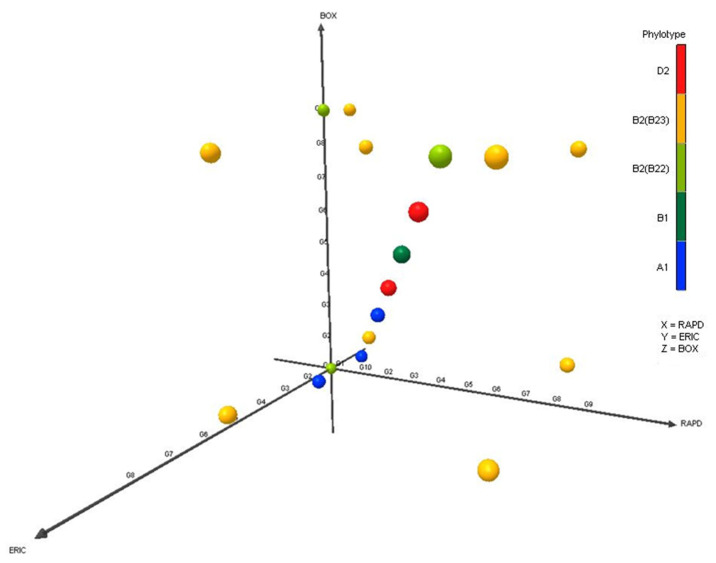
The diversity of APEC isolates according to various typing systems. The principle component analysis (PCA) plots represent the distribution of different phylogroups. The PCA plot is represented by three (X, Y, and Z axes) dimensional orientation, where different color codes indicate respective (orange for phylogroup D2; blue, for A1, yellow for B2_3_, green for B2_2_, and dark red green for B1) phylogroups in the PCA plots. Most of the samples of the corresponding phylogroup clustered in the first quadrant, indicating their close phylogenetic relationship.

**Figure 3 microorganisms-08-01135-f003:**
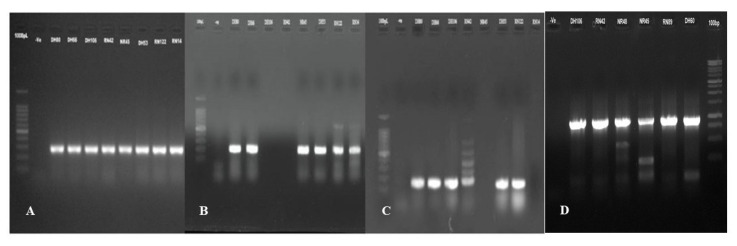
Representative PCR results for the detection of APEC phylogenetic groups among the colibacillosis cases of Bangladeshi poultry samples. Here, (**A**): (*chu*A, 279bp), (**B**): (*yja*A, 211bp), (**C**): (tspE4.C2, 152bp), (**D**): (*arp*A, 400bp), and Lane 1 is molecular ladders (100bp), Lane 2 is negative blank control, and Lanes 3–10 are the strains DH80, DH66, DH106, RN42, NR45, DH53, RN122, and RN14, respectively. DNA bands at the appropriate position were observed in APEC strains DH66, DH80, DH53, and RN122, denoted as phylogroup B2(B2_3_) (*chu*A+, *yja*A+, *arp*A-, tspE4.C2+), APEC strains DH106, RN42 represented as phylogroup D2 (*chu*A+, *yja*A-, *arp*A-, tspE4.C2+), and APEC strains NR45, RN14 denoted as phylogroup B2(B2_2_) (*chu*A+, *arp*A+, *yja*A+, tspE4.C2-).

**Figure 4 microorganisms-08-01135-f004:**
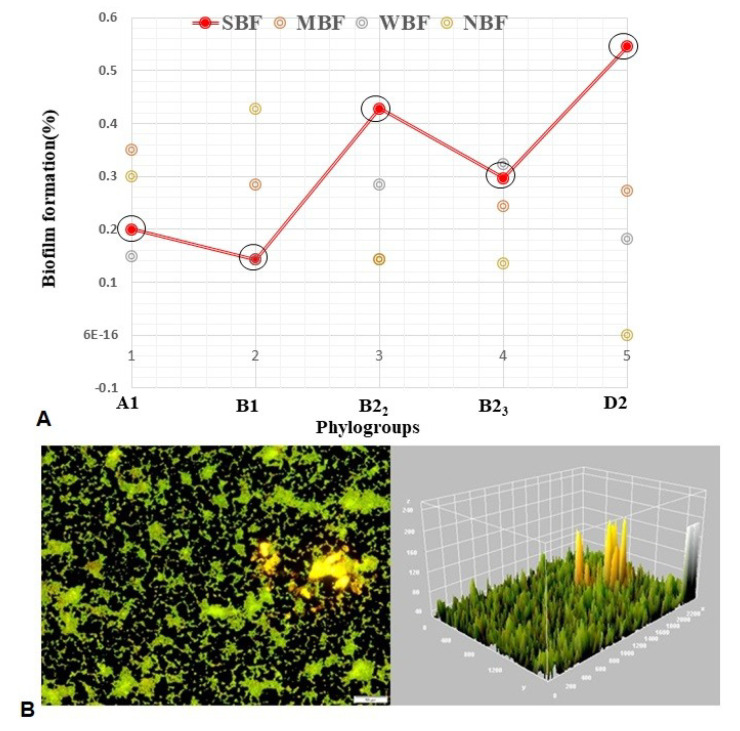
Biofilm formation (BF) ability of the APEC phylogroups. (**A**) Diagrammatic representation of BF in various phylotypes. Here: SBF, strong biofilm formers; MBF, moderate biofilm formers; WBF, weak biofilm formers; NBF, non-biofilm formers. Red solid line with black circle represented the SBF ability fluctuations in which the isolates of phylogroup B1 had the lowest (14%) BF ability, and the phylogroup D2 isolates showed the highest (54.55%) BF ability. (**B**) Fluorescence microscopy images of isolate (RN3, D2) under 20× magnification. Biofilm stained with film tracer LIVE/DEAD biofilm viability kit. Live or active cells are fluorescent green and dead or inactive cells are fluorescent red. Surface plot of 3D volume image (center image) and cross section of 3D volume image (right side image) show the distribution of live and dead cells throughout biofilm layers.

**Figure 5 microorganisms-08-01135-f005:**
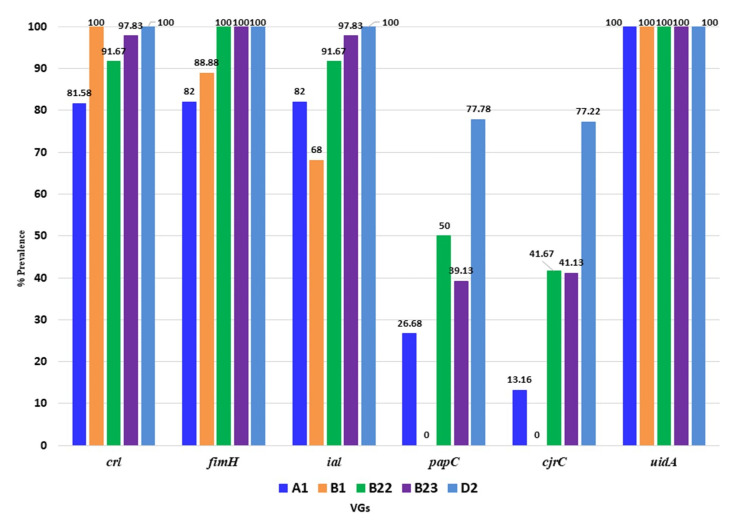
Prevalence of the three pathogenic genes among the APEC phylotypes. Here *x*-axis represents the phylotypes, and the *y*-axis denotes the prevalence (%) of the genes among the *E. coli* isolates. For phylotype A1: First column represents the *crl* gene; while second for *fim*H; third for *ial*; fourth for *pap*C, fifth for *cjr*C, and sixth for *uid*A gene. This serial is true for all the other phylotypes.

**Figure 6 microorganisms-08-01135-f006:**
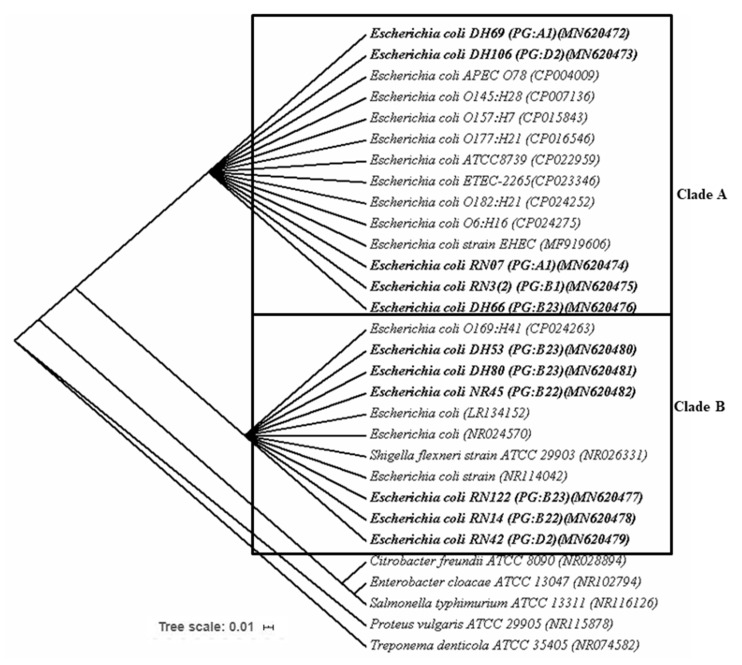
Phylogenetic tree predicted by the neighbor-joining method using 16S rRNA gene sequences. The evolutionary distances were computed using the Kimura two-parameter model method. The bootstrap considered 1000 replicates. The scale bar represents the expected number of substitutions averaged over all the analyzed sites. The optimal tree with the sum of branch length = 0.35475560 is shown here. *Treponema denticola* was used as out group. The length of the scale bar represents one nucleotide substitution per 100 positions.

**Figure 7 microorganisms-08-01135-f007:**
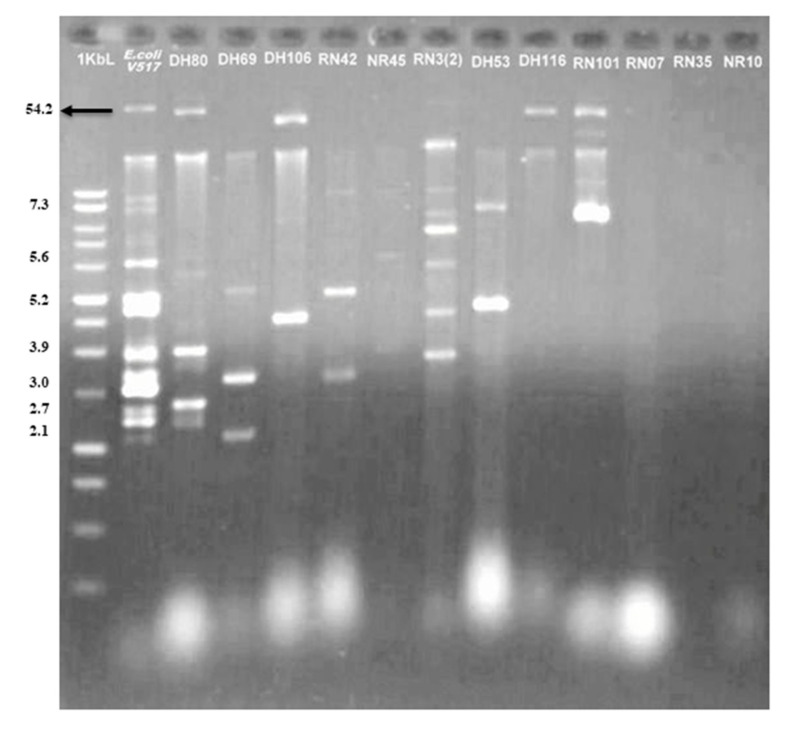
Plasmid profile of representative multidrug-resistant (MDR) APEC isolates. Agarose gel electrophoretogram (0.8% gel) of plasmid DNA of the MDR isolates: APEC strain DH80, DH69, DH106, RN42, NR45, RN 3(2), DH53, DH116, RN07, RN35, NR10 (from lane 3 to lane 14), *E. coli* V517 in lane 2 was used as the marker, and its plasmids of different sizes (kb) have been denoted in the figure. Lane 1 is molecular ladder. Plasmid bands at the same position was observed in APEC strain DH80, RN3(2) at the position between 3.9 kb; strain DH69, RN42 at the positions 3.0 kb, strain DH53, RN101 at the position of about 7.3 kb, and strain DH106, DH53, RN3(2) at the positions between 5.2 to 3.0 kb. Large plasmids at common positions were observed in strain DH80, DH106, DH116, RN101. Strain RN07, RN35, NR10 did not harbor any plasmid.

**Figure 8 microorganisms-08-01135-f008:**
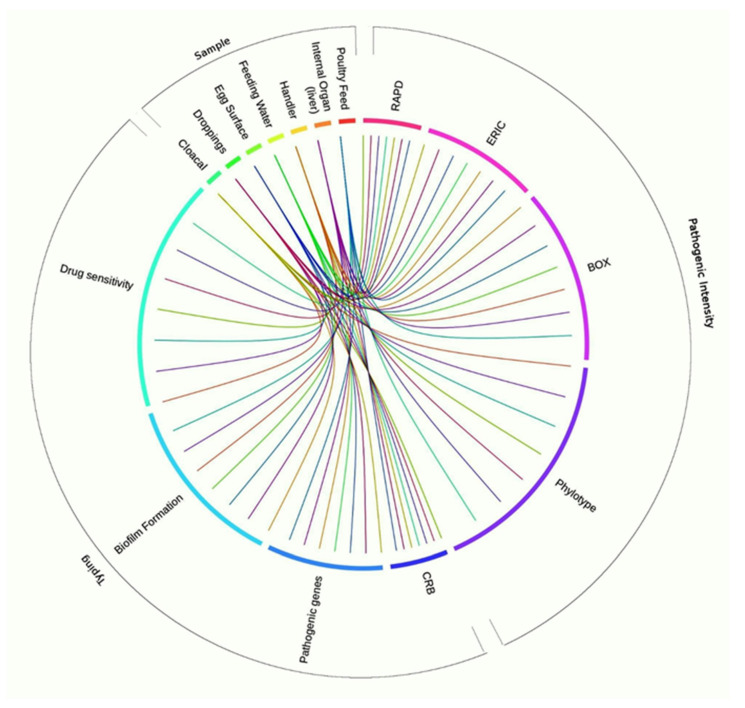
Circos plot representation of association of different sample types with molecular typing and pathogenic intensity characterization of *E. coli*. The association showed significant (*p* = 0.032) correlation. The frequency of occurrence of different features, such as sample categories, typing, and pathogenicity patterns, is depicted in the outer ring. The inner ring of Circos plot depicts the correlation between the sample categories (cloacal, droppings, egg surface, feeding water, handler, internal organ, feed) and molecular typing (RAPD, ERIC, BOX, phylotype) and pathogenic intensities (CRB, pathogenic genes, biofilm formation, drug sensitivity). Each factor has been assigned to a color. The arc originates from sample types and terminates at typing and pathogenic intensity levels to compare the association between the origin and terminating factors. The area of each colored ribbon depicts the frequency of the samples related with the particular typing and pathogenic intensity expression.

**Figure 9 microorganisms-08-01135-f009:**
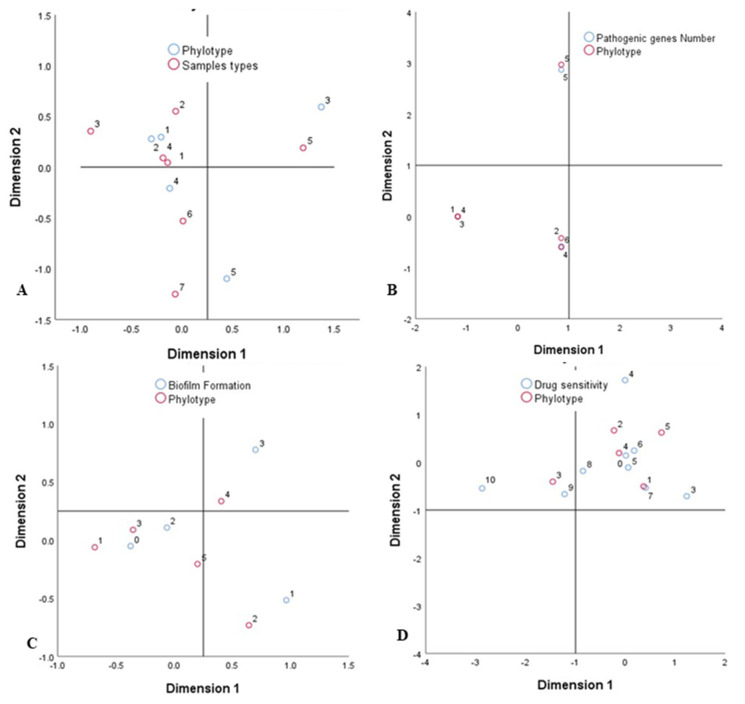
Correspondence analyses (CA) for the categorical variables. Sample types, pathogenicity, and phylogroups that are similar where this two-dimensional representation explain 100% of the total variation, with 68.55% explained by the first dimension, and 31.45% by the second dimension. On the other hand, for the drug sensitivity and biofilm formation, the representation explains discriminatory variations by the respective dimension. Here (**A**) represents phylotypes (white circle) vs sample types relation (red circle); (**B**) represents phylotypes (red circle) vs pathogenic gees relation (white circle); (**C**) represents phylotypes (red circle) vs biofilm formation (white circle); (**D**) represents phylotypes (red circle) vs drug sensitivity (white circle).

**Table 1 microorganisms-08-01135-t001:** Relationship between APEC phylotypes and drug sensitivity.

Resistance % (Strains)						
**Antibiotics**	% (Total)	A1 (*n* = 40)	D2 (*n* = 11)	B2_3_ (*n* = 43)	B2_2_ (*n* = 11)	B1 (*n* = 9)
**Ampicillin (AMP)**	74.56 (84)	70 (28)	63.64 (7)	74.42 (31)	90.91 (10)	88.89 (8)
**Doxycycline (Do)**	78.95 (90)	77.5 (31)	72.73 (8)	79.07 (34)	72.72 (8)	100 (9)
**Tetracycline (Te)**	75.44 (86)	85 (34)	90.91 (10)	83.72 (36)	81.82 (9)	77.78 (7)
**Nitrofurantoin (F)**	63.16 (72)	60 (24)	63.64 (7)	65.12 (28)	72.72 (8)	55.56 (5)
**Ciprofloxacin (CIP)**	44.74 (51)	35 (14)	45.46 (5)	51.16 (22)	45.45 (5)	44.45 (4)
**Nalidixic acid (NA)**	76.32 (87)	65 (26)	90.91 (10)	76.74 (33)	90.11 (10)	88.89 (8)
**Cefoxitin(Fox)**	41.23 (47)	45 (18)	9.01 (1)	50 (21)	27.27 (3)	44.45 (4)
**Imipenem (IMP)**	22.81 (26)	17.5 (7)	18.18 (2)	16.28 (7)	36.36 (4)	66.65 (6)
**Gentamycin (GN)**	26.32 (30)	17.5 (7)	27.27 (3)	23.26 (10)	45.45 (5)	44.46 (4)
**Chloramphenicol (C)**	51.75 (59)	42.5 (17)	54.55 (6)	60.47 (26)	45.45 (5)	66.65 (6)
**Sulfonamide (S)**	44.74 (51)	45 (17)	45.46 (5)	48.84 (21)	27.27 (4)	55.56 (5)
**Azithromycin (ATM)**	31.58 (36)	27.5 (11)	30.23 (13)	30.23 (13)	54.55 (6)	33.36 (3)
**Polymexin B (Pb)**	7.89 (9)	7.5 (3)	18.18 (2)	9.30 (4)	0 (0)	0 (0)

Here, *n* = total tested isolates; A1/D2/B2_3_/B2_2_/B1 = the APEC phylotyes.

## Data Availability

The 16S rRNA gene sequencing data (11 sequences) has been submitted to NCBI database under the accession numbers-MN620472-MN620482.

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
