# Peer review of "Multidrug-Resistant Avian Pathogenic Escherichia coli Strains and Association of Their Virulence Genes in Bangladesh"

_microorganisms, 2020, doi:10.3390/microorganisms8081135_

Round 1
Reviewer 1 Report
The manuscript is analyzing the etiology of the avian pathogenic Escherichia coli (APEC). Since this bacterium is causing colibacillosis worldwide, the manuscript is relevant and interesting from the epidemiological point of view. There are some errors and mistakes that should be corrected.
- Spelling and grammar errors all over the text. Example: line 57, the first sentence in Introduction.
- 2.2 section needs a complete composition of the used nutrient broth. what exact media was used for the cultivation of bacteria and inoculation before plating on MacConcey agar?
- 2.3.3 it should be stated what general magnification (100x, 1000x?) was used for fluorescent microscopy.
- What exactly is "molecular biology-grade water"? distilled water?
- 2.4 section is not clear (even after reading the reference No 50) how the cell lysis is performed in this method? Was there any cell lysis step at all? Whats was the amount of the extracted genomic DNA?
- Check carefully the typing errors. There are no spaces between numbers and dimensions in many places of the manuscript.
After all, in my opinion, the article could be shorter. Since the exact amount of the samples was taken from the certain farms it is not necessary to mention all the percentages at the 3.1 section.
Author Response
Reviewer # 1
Reviewer’s comment (RC)
Spelling and grammar errors all over the text. Example: line 57, the first sentence in Introduction.
Authors’ response (AR):
We appreciate the knowledgeable reviewer for the encouraging comments and suggestion for improvement of the manuscript. We agree with the reviewer and tried to comprehensively revise the entire manuscript. We’ve deleted the double words from the line 58 followed by correcting all spelling and grammar errors, and revised manuscript accordingly. Please go through the revised manuscript (Line 58).
Reviewer’s comment (RC)
2.2 section needs a complete composition of the used nutrient broth. what exact media was used for the cultivation of bacteria and inoculation before plating on MacConcey agar?
Authors’ response (AR):
Thank you for this valuable comments. We added the complete composition of the used nutrient broth used for the cultivation of bacteria before plating on selective agar media (MacConcey and Eosin Methylene Blue agars). We’ve revised the manuscript accordingly. Please see lines 142-146 in the revised manuscript.
Reviewer’s comment (RC)
2.3.3 it should be stated what general magnification (100x, 1000x?) was used for fluorescent microscopy.
Authors’ response (AR):
We would like to thanks the reviewer. We used a general magnification of 40X objective for fluorescence microscopy, and the literature is added in the revised manuscript. Please see lines 180-181 in the revised manuscript.
Reviewer’s comment (RC)
What exactly is "molecular biology-grade water"? distilled water?
Authors’ response (AR):
We would like to thank the reviewer. We used filtered distilled water to resuspend the DNA for final extraction. We’ve revised accordingly. You cordially requested to go through lines 186-190 in the revised manuscript.
Reviewer’s comment (RC)
2.4 section is not clear (even after reading the reference No 50) how the cell lysis is performed in this method? Was there any cell lysis step at all? Whats was the amount of the extracted genomic DNA?
Authors’ response (AR):
We appreciate the reviewer for raising a critical issue. Actually, we performed the simple heat boiled method of DNA extraction which includes some physical factors or steps such as heating, cooling, freezing, and high speed beating using ultracentrifuges (Skowronski et al.., 2000; Agarwal et al., 2001; Zhu et al., 2005; Jose et al., 2006). In this methods no chemicals like commercial cell lysate are needed. High temperature exposure (100°C) is known to cause damage to cell membranes and cell walls (Jose et al., 2006; Höfler et al., 1994). Jose and Brahmadathan reported that heating at 94 °C for two minutes was enough to denature cell walls (Jose et al., 2006). Low temperatures were also observed to destroy cell walls and membranes. Freezing induces crystallization of water inside cells which leads to destruction of cytoplasmic structures (Jose et al., 2006; Tell t al., 2003). High speed beating using ultracentrifuges also helps to destruction of the bacterial cell wall and membranes (Dashti et al., 2009). All of these steps were employed in our boiled DNA extraction method. We’ve added the amount of the extracted genomic DNA (120-150 microliter per sample) in the revised manuscript.
Please see lines 184-191 in the revised manuscript.
References
Dashti, A. A., Jadaon, M. M., Abdulsamad, A. M., & Dashti, H. M. (2009). Heat treatment of bacteria: a simple method of DNA extraction for molecular techniques. Kuwait Med J, 41(2), 117-122.
Tell LA, Foley J, Needham ML, Walker RL. Comparison of four rapid DNA extraction techniques for conventional polymerase chain reaction testing of three Mycobacterium spp. that affect birds. Avian Dis 2003; 47:1486-1490.
Skowronski EW, Armstrong N, Andersen G, Macht M, McCready PM. Magnetic, microplate-format plasmid isolation protocol for high-yield, sequencing-grade DNA. Biotechniques 2000; 29:786-788.
Agarwal A, Kumar C, Goel R. Rapid extraction of DNA from diverse soils by guanidine thiocyanate method. Indian J Exp Biol 2001; 39:906-910.
Zhu K, Jin H, Ma Y, et al. A continuous thermal lysis procedure for the large-scale preparation of plasmid DNA. J Biotechnol 2005; 118:257-264.
Jose JJ, Brahmadathan KN. Evaluation of simplified DNA extraction methods for emm typing of group A streptococci. Indian J Med Microbiol 2006; 24:127-130.
Höfler G. Detection of bacterial DNA using thepolymerase chain reaction (PCR). Verh Dtsch Ges Pathol 1994; 78:104-110.
Reviewer’s comment (RC)
Check carefully the typing errors. There are no spaces between numbers and dimensions in many places of the manuscript.
Authors’ response (AR):
We would like to thank the reviewer. We are sorry for the spelling and grammar errors. We’ve revised the manuscript, and please go through the revised manuscript.
Reviewer’s comment (RC)
After all, in my opinion, the article could be shorter. Since the exact amount of the samples was taken from the certain farms it is not necessary to mention all the percentages at the 3.1 section. Authors’ response (AR):
Thank you for the encouraging comment. We have revised the 3.1 section accordingly. Please go through Lines 298-306 in the revised manuscript.
Reviewer 2 Report
In the present study, the authors have examined ~400 avian pathogenic Escherichia coli (APEC) strains which are responsible for most of the cases of avian colibacillosis in Bangladesh. The APEC isolates were phylogenetically analyzed and classified into five phylotypes. Moreover, these strains have been screened for the presence of virulence genes and for their resistance to antimicrobial agents. Importantly, this study identifies B2 and D2 phylotypes as the most virulent and very often they are multidrug resistant. Generally, the authors made a great deal of work although it was quite routine. They used basic microbiological and molecular biology procedures. Anyhow, experiments shown are quite well designed and results support the main conclusions drawn by the authors.
Several points should be addressed.
1) A general concern is that the present study is very specific and results obtained are strongly related with the microbiological/epidemiologic and social-economic situations of Bangladesh. Although a comparative analysis could be done, I would have some doubts about the possible use of these data in other countries.
2) The genetic diversity estimated by RAPD, ERIC PCR and BOX PCR results in 10, 8 and 9 groups, respectively. It is not clear what “group 1-4 clustered in the same quadrant of the PCA plot” (line 306) is referred to. Differently, the phylogroups shown in Fig. 2 are five. This point should be explained.
3) Letters to indicate panels are missing in Fig. 3.
4) The sentence “Solid line with circle represented the SBF ability fluctuation between the phylotypes” (legend of Fig. 4, lines 357-358) is confusing. From Fig.4A, it seems that the circles indicate the strongest BF for each phylogroup but this is not valid for A1 and B1. Strangely, the solid line does connect all the circles and so it not clear what line and circles represent, respectively. The plot of Fig. 4A deserves some explanations.
5) It is not clear the association between the two virulence genes, possibly parC and cjrC (they should be indicated), and types B2 and D2 (line 379). One could conclude that parC and cjrC, on average, are less abundant in all phylotypes with the exception of D2 (~78 % in Fig. 5).
6) The reference for plasmids carried by V517 strains should be added (DOI: 10.1016/0147-619x(78)90056-2).
7) In my opinion, the article is not well written and a lot of sentences are difficult to understand. In addition, the Discussion section is too long. Several grammar and spelling mistakes have to be corrected. For example: the word “ubiquitous” is duplicated (line 57); phylogrpoups in the legend of Fig. 2 (line 309); “fluctuation between the phylotypes” is wrong (line 358), it should be ”among phylotypes”; line 386, ”…all the others phylotypes” ; line 543, “….din not”; line 569 “…higher than those resistant ones”; line 578, “whether the mere acquisition….. or if a specific”.
Author Response
Reviewer ## 2
Reviewer’s comment (RC)
A general concern is that the present study is very specific and results obtained are strongly related with the microbiological/epidemiologic and social-economic situations of Bangladesh. Although a comparative analysis could be done, I would have some doubts about the possible use of these data in other countries.
Authors’ response (AR):
Thank you for the encouraging comment. Yes, we understood that the scope of the manuscript is broad, and will impart new messages for the academician, researchers and field veterinarians regarding APEC and for its containment in the poultry settings. We are happy to carry out a comparative analysis in coming future following the reviewer’s suggestion. Moreover, we hope and do believe that this method of APEC isolation, phylotyping, virulence and antimicrobial assays could be reproducible anywhere in the world including our neighboring countries.
Reviewer’s comment (RC)
The genetic diversity estimated by RAPD, ERIC PCR and BOX PCR results in 10, 8 and 9 groups, respectively. It is not clear what “group 1-4 clustered in the same quadrant of the PCA plot” (line 306) is referred to. Differently, the phylogroups shown in Fig. 2 are five. This point should be explained.
Authors’ response (AR):
Thank you for pointing out the weakness of our manuscript. We happily revised the manuscript and legend of Figure 2. The PCA plot is represented by three (X, Y and Z axes) dimensional orientation where different color codes indicate respective (Orange for phylogroup D2, blue for A1, Yellow for B23, Green for B22, and dark red green for B1) pyhlogroups. Most of samples of the corresponding phylogroups clustered in the first quadrant indicating their close phylogenetic relationship. Please go through Lines 328-334 in the revised manuscript.
Reviewer’s comment (RC)
Letters to indicate panels are missing in Fig. 3.
Authors’ response (AR):
We would like to thank the reviewer for pointing out an import error. We’ve revised the Figure 3 with addition of letter panels for the respective band lengths, and the figure legend has also been revised. Please see the Figure 3 and its legend in the revised manuscript. Please see Lines 354-362, and Figure 3 in the revised manuscript.
Reviewer’s comment (RC)
The sentence “Solid line with circle represented the SBF ability fluctuation between the phylotypes” (legend of Fig. 4, lines 357-358) is confusing. From Fig.4A, it seems that the circles indicate the strongest BF for each phylogroup but this is not valid for A1 and B1. Strangely, the solid line does connect all the circles and so it not clear what line and circles represent, respectively. The plot of Fig. 4A deserves some explanations.
Authors’ response (AR):
We would like to thank the reviewer for his scholastic review. We’ve revised the figure as well as the figure description in the revised manuscript. Please see Lines 384-389 in the revised manuscript.
Reviewer’s comment (RC)
It is not clear the association between the two virulence genes, possibly parC and cjrC (they should be indicated), and types B2 and D2 (line 379). One could conclude that parC and cjrC, on average, are less abundant in all phylotypes with the exception of D2 (~78 % in Fig. 5).
Authors’ response (AR):
Thank you very much for your understanding and nice comments. We’ve revised the manuscripts accordingly. Please go through Lines 408-412 in the revised manuscript.
Reviewer’s comment (RC)
The reference for plasmids carried by V517 strains should be added (DOI: 10.1016/0147-619x(78)90056-2).
Authors’ response (AR):
We would like to thank the reviewer. We’ve added the suggested reference, and revised the reference section also. Please go through the results section (lines 268, 938-940) of the revised manuscript).
Reviewer’s comment (RC)
In my opinion, the article is not well written and a lot of sentences are difficult to understand. In addition, the Discussion section is too long. Several grammar and spelling mistakes have to be corrected. For example: the word “ubiquitous” is duplicated (line 57); phylogrpoups in the legend of Fig. 2 (line 309); “fluctuation between the phylotypes” is wrong (line 358), it should be ”among phylotypes”; line 386, ”…all the others phylotypes” ; line 543, “….din not”; line 569 “…higher than those resistant ones”; line 578, “whether the mere acquisition….. or if a specific”.
Authors’ response (AR):
We would like to thank the reviewer for nice and inquisitive look into the entire manuscript, and pointing out several minor errors throughout the manuscript. We’ve revised and edited the whole manuscript accordingly. Please go through the revised manuscript ( highlighted).